# Network and Exploratory Factorial Analysis of the Depression Clinical Evaluation Test

**DOI:** 10.3390/ijerph191710788

**Published:** 2022-08-30

**Authors:** María Guillot-Valdés, Alejandro Guillén-Riquelme, Juan Carlos Sierra, Gualberto Buela-Casal

**Affiliations:** 1Mind, Brain and Behavior Research Center, University of Granada, 18011 Granada, Spain; 2Faculty of Health Sciences, Valentian International University, 46002 Valencia, Spain

**Keywords:** depression, assessment, factor analysis, network analysis, psychometrics

## Abstract

Depression is a highly prevalent disorder with a wide range of symptomatology. Existing instruments for its assessment have only a few items for each factor. The Depression Clinical Evaluation Test (DCET) has been created to cover all depression symptoms at different times (month, year, and always) with several items for each facet. The content validity of this instrument has been judged by experts and, in this paper, we analyse its factorial structure and make a network analysis of it. The test (196 items) was administered to 602 adults without psychological disorders (*M*_age_ = 24.7, SD = 8.38, 72% women) both online and on paper. A network was estimated for each time point, using the absolute minimum selection and shrinkage operator. From the factor analysis, 12 factors were established for month, 11 for year, and 10 for always, leaving 94 items. The network analysis showed that the facets of depressive mood, anhedonia, and thoughts of Death, are central to all the estimated networks. The DCET is proposed as a valid and reliable multifactorial instrument to detect the variability of depressive symptoms in adults, guaranteeing its diagnostic usefulness.

## 1. Introduction

Depression is a mental disorder characterized by a decline in mood and loss of interest or pleasure in activities that the person normally enjoyed, among other somatic and cognitive symptoms, causing clinically significant distress or impairment in social, occupational, or other important areas of functioning [1]. According to World Health Organization estimates, more than 350 million people worldwide (4.2%) suffer from depression [2], posing a public health risk and even a danger of consummated suicide, when there is residual symptomatology [3,4].

Regarding the assessment of depression symptoms, it is necessary to have instruments with adequate psychometric properties. However, it is also necessary to ensure their diagnostic utility and clinical discrimination [5,6]. One of the drawbacks lies in the fact that major depression is a disorder with a broad symptomatology that includes cognitive, motor, behavioural, and somatic symptoms, in addition to the core emotional symptoms of the disorder. In the assessment of depression, most instruments are focused solely on the emotional factor in adults, children, and adolescents [7]. Moreover, those instruments that assess several areas do so with only a few items for each of the dimensions [8]. Thus, for example, among the most widely used assessment instruments [9] are the Beck Depression Inventory [10] and the Zung Depression Scale [11], which assess most of the symptoms of depression, but with a reduced number of items, and the Hospital Anxiety and Depression Scale [12] focused on emotional factors.

Another aspect that hinders the assessment of depression is the debate on the dimensional or unitary vs. categorical perspective, which can be generalised to psychopathology in general. The dimensional approach advocates a broad coverage of disorders, with a more holistic approach. Thus, from this perspective, the development of instruments with various items for each symptom that allow the collection of multiple manifestations of its occurrence is preferable [13,14,15,16].

Taking into account all these considerations and following the standards of the American Educational Research Association et al. [17], a theoretical approach for an expert-validated depression test was developed to ensure its construct and content validity, and the content of an item bank based on this theoretical model was validated [18]. From these two procedures, we obtained the Depression Clinical Evaluation Test [18], which covers all manifestations of depressive symptomatology with several items per factor, and assesses several temporal moments. As a continuation of this first study of the Depression Clinical Evaluation Test (DCET), we propose to provide evidence based on its internal structure and to carry out its network analysis.

In this network model, disorders are conceived as a complex and dynamic system, in which symptoms (nodes) are interrelated and connected by edges [19]. Several works have used this model to analyse different disorders: depression and anxiety [20,21,22] or narcissism personality [23].

Therefore, this paper aims to: (1) examine the differential item functioning in two assessment procedures (online vs. paper and pencil), (2) examine the factor structure of the DCET, (3) analyse its reliability, and (4) perform a network analysis of the final factors comprising the DCET.

## 2. Materials and Methods

### 2.1. Participants

The sample consisted of 602 adults aged 18 to 85 years from different geographical areas of Spain (*M*_age_ = 24.7, SD = 8.38, 72% women). Of the total, 500 were evaluated online (see Appendix A). This sample size coincides with the required sample size by using the online calculator of QuestionPro, setting the confidence interval at 95%, the margin error at 3.99, and a study population of 36,000,000 (Spanish people older than 18) and giving a result of 604. This is based on the following formula, recommended by some authors [24], where Z represents the punctuation associated with the confidence interval (95%), *p* = 0.5 and c would be the margin error, in this case, 3.99:Sample size = Z2 × (p) × (1 − p)/c2(1)

The remainder answered the survey in paper-and-pencil format. The socio-demographic data of the participants are shown in Table 1.

### 2.2. Instrument

The Depression Clinical Evaluation Test [18] comprises 196 items with a Likert-type multiple response format from 0 (*Almost never*) to 4 points (*Completely*), to be answered for three time points (last month, last year, and always), allowing the obtaining of information on the symptom at present, in the previous year, or throughout life. The test includes behavioural symptoms (e.g., ‘I have been drinking alcoholic beverages every time I go out’), affective symptoms (e.g., ‘I feel sad’), somatic symptoms (e.g., ‘I wake up at night and find it hard to fall sleep again’), cognitive symptoms (e.g., ‘It is hard for me to keep my attention for a long time’), and interpersonal symptoms (e.g., ‘I spend less time with my partner’).

This instrument was created from a bank of 300 items that was subjected to a qualitative evaluation by 13 PhD experts in clinical psychology, psychometry, and/or psychopathology. They had experience in the subject due to their academic training and work experience. They were gathered from different Spanish institutions/universities and contacted by email, explaining the purpose of the project and asking for their collaboration. The degree of comprehension of each item was analysed in a sample of 50 adults. The experts evaluated the items based on the criteria of content, relevance, clarity, comprehension, sensitivity, and offensiveness. This process involved the elimination of a total of 104 items and the premise that the instrument had adequate construct and content validity [18].

### 2.3. Procedure

The test was carried out in two ways: (a) online electronic form (answered by 500 people) through the Google Forms system, contacting participants through social networks (Facebook, Twitter, and Instagram) by using the snowball procedure, and dissemination of emails from the University of Granada; and (b) face-to-face (102 people of the total) in public places (university centres, train, and bus stations). In the online form, the first page of the questionnaire included detailed information on the study, data protection guarantees in accordance with current regulations, and informed consent. All participants had to respond affirmatively to the contents of this sheet before moving on to the questions of the rest of the questionnaire. The answering of several questionnaires from the same device was prevented by registering the IP, which eliminated duplicate questionnaires. In the same way, automatic response patterns were analysed manually to eliminate this type of case, and no anomalous response pattern was found.

The paper administration was carried out by a single researcher, who went to places with a high presence of people (e.g., bus and train stations). There she indicated to the participants the object of the study, as well as the ethical guarantees provided. Once informed consent was given and the confidentiality clause signed, the investigator was present during the holding of the tests to resolve any doubts that could arise.

In both procedures, to guarantee the confidentiality of the responses, the questionnaires were answered anonymously. No incentives were offered and there were no refusals to participate in the study. The estimated time for completion was 25 min. The study was approved by the Human Research Ethics Committee of the University of Granada with reference 2576/CEIH/2022 and the ethical standards of the Declaration of Helsinki [25] and its subsequent amendments were guaranteed.

### 2.4. Data Analysis

First, a differential item functioning analysis (DIF) of the DCET measure was conducted based on the administration mode (online vs. paper). The presence of DIF implied that the response to an item is influenced by group characteristics and not only by the level of the construct [26,27]. The mean scores of the DCET factors were compared according to the application modality (paper vs. online). The three-stage logistic regression procedure was used, which is detailed in Guillot-Valdés et al. [5]. Three models were compared; 2-1 indicated a uniform DIF and 3-2 indicated a non-uniform DIF. To find evidence of the internal structure of the scale, an exploratory factor analysis (EFA) was performed, following the MINRES method on the matrix of polychoric correlations and oblimin rotation, since a correlation between the factors is assumed. The Barlett sphericity and Kaiser–Meyer–Olkin (KMO) test assumptions were tested and were found to be adequate. To determine the initial number of factors, the *R* software package, *nfactors,* was used to estimate the various indicators of the retention of the number of factors (i.e., Sample Size Adjusted, Parallel Analysis, Minimum Average, and Partial Very Simple Structure). The factorisation was checked to see if there were factors without clear loadings on any of the items or if the saturated items lacked theoretical sense. The number of factors was reduced following this process of reducing the number of factors and items without clear or correct saturations in their theoretical aspects. This process was repeated until a stable solution was achieved, where the remaining items were saturated on the appropriate factor. Factor loadings of 0.30 were considered as the minimum criterion to allow the membership of an item on its factor. In the case of an item saturating on two factors, it was retained if the saturation was correct on its factor and the difference was less than 0.15. Sub-factors were only unified if they belonged to the same theoretical factor of the DCET. This process was performed for the responses pertaining to the *month* temporal moment. Once an adequate and stable factorisation was found, the same number of factors was initially used for the *year* and *always* categories as well.

To examine the internal consistency of the instrument, the ordinal alphas were calculated for all the temporal moments and factors.

### 2.5. Network Analysis

A weighted and undirected network was estimated. The ‘nodes’ of the network corresponded to the factors of the questionnaire and the ‘edges’ were the correlations between them [28]. The Fruchterman–Reingold algorithm was used for the network design, which places the most important nodes in the centre of the network and the weakest ones on the periphery [29]. In the network, the thickness of the lines indicates the strength of the associations between nodes. The colour of the line indicates the direction of the association (blue lines indicate positive associations; red lines indicate negative associations). Factors corresponding to each time point were included and grouped according to their theoretical factors, each with a colour (Figure 1): Affective (depression, anhedonia, and guilt), Cognitive (attention and death), Behavioural (substances), Interpersonal (family, distress, and partner) and Somatic (appetite, sleep, libido, and fatigue). In order to minimize the number of spurious relationships, network models were estimated using a least absolute shrinkage and selection operator (LASSO), a regularisation algorithm that sets partial correlations to zero and requires fewer parameters to estimate [30]. The function ‘estimateNetwork’ was estimated with *bootnet*, using ‘EBICglasso’ as the default method and 0.5 as the fitting parameter. To see the importance of each node in the network, the *degree centrality*, which is the number of connections that a node has with the others, and the *expected influence*, which is the sum of all the edges of a node, were calculated.

To estimate network stability, a nonparametric *bootstrap* analysis was also performed, following the procedure of Liu et al. [22]. This showed us the estimated precision of the ‘edges’ and the centrality indices [31]. To calculate the 95% confidence intervals (CI) of the edge values and to estimate the stability of the centrality indices, 600 permutations were performed, using a case-dropping subset bootstrapping. This method was used for the item set of *month, year,* and *always.*

Analyses were performed with the statistical programme R 3.5.1 [32] and JASP Team software [33]. The following R packages were used: *psych* [34], *nFactors* [35], *mvtnorm* [36], *car* [37], *tibble* [38], *bootnet* [31], and *Qgraph* [39].

## 3. Results

### 3.1. Differential Item Functioning (DIF)

Since the sampling was undertaken in two different formats, the first step was to confirm the absence of bias between the two applications (paper and pencil). As no serious DIF problems were observed, we proceeded to unify both samples.

In the case of the comparison for the temporal moment of month, in model 2-1 no DIF was observed. All χ2 ranged between 0 and 0.98, with a significance greater than 0.01. The same (χ2 values between 0.002 and 0.99, *p* > 0.01) was seen for model 3 vs. 2, except for item 86 (‘I have set a time and day to commit suicide’), in which a moderate DIF was observed: Δ χ2 < 0.01, Δ *p* = 0.993, Δ Nagelkerke R^2^ = 0.052. A value of Δ Nagelkerke R^2^ lower than 0.035 would indicate an inappreciable DIF from 0.035 to 0.07 or a moderate DIF, and higher than 0.07 would be a high DIF [40].

For the temporal moment year, for both models 2-1 and 3-2, the χ2 ranged from 0 to 0.99, with a significance level greater than 0.01, and no DIF was found (Δ Nagelkerke R^2^ < 0.035 in the two models).

For the temporal moment always, for both models 2-1 and 3-2, the χ2 ranged between 0 and 1. The presence of DIF was also not found.

### 3.2. Exploratory Factorial Analysis (EFA)

The EFA was conducted with all items (196) and all study participants (N = 602). The KMO test result was 0.87 for the temporal moment month, 0.88 for year, and 0.84 for always. The results of Bartlett’s test of sphericity were statistically significant (*p* < 0.001) in all cases. Four factorisations were made for the temporal moments, month and always, and six for year, leaving a total of 94 items, 12 factors for the time moment month (explaining 67% of the variance), 11 for year (70% of the variance), and 10 for always (explaining 65% of the variance). The name and content of each of the factors were determined after factorisation. For this purpose, the theoretical factor of the items with the highest saturations in the factor obtained in the analysis was analysed. This process implied excluding those items that did not show high saturations (lower than 0.30) or the ones that saturated highly in factors, which, theoretically, should not saturate. In the case of year, the union of two factors (depression and anhedonia) was observed, leaving one with no explanatory load. At the temporal moment always, the factor Libido disappeared, Couple was merged with Family and the factor Fatigue was formed, but it was eliminated for the final version of the questionnaire. Appendix B shows the saturations of the items, the proportion of variance explained by each factor, and the commonalities at each of the temporal moments (month, year, and always).

Finally, the distribution of items was as follows: decreased attention/poorer task performance (20 items), thoughts of death (19), depressed mood (12), anhedonia (6), sleep disturbance (6), malaise (6), appetite disturbance (5), family impairment (5) and couple impairment (4), thoughts of undervaluation and guilt (5), substance abuse (3), and decreased libido (3).

Owing to different factorisations depending on the temporal moment, the corrections for some of the factors vary slightly according to each factor.

### 3.3. Reliability

Internal consistency was estimated for each of the factors obtained and for each of the temporal moments, while the ordinal alphas ranged from 0.75 to 0.90.

### 3.4. Network Analysis

The facets were grouped according to the factor to which they belong, as can be seen in Figure 1. In this case, the affective factor grouped the facets of depressed mood, anhedonia, and guilt, as do the somatic factor (libido, sleep, and appetite loss). However, this was not the case with the interpersonal and cognitive factors, whose facets were more dispersed in the network. It should be noted that all nodes established positive connections with each other, except in the always network, where there were negative relationships between the depressive mood and attention nodes, and between guilt and attention (Appendix C). Thus, in the case of month and year, the expected influence values were identical to those of the strength of centrality; so, these are not shown. As can be seen in Figure 1 (network corresponding to the temporal moment month), the most central nodes were depressive mood, anhedonia, and death thoughts. Some facets had stronger connections, such as depressed mood and anhedonia, or guilt and thoughts of death. These nodes also had a higher degree index, and, in terms of the expected influence, they were the most influential (Figure 2), while the facets of substance abuse such as decreased libido and clinical discomfort had lower indices and were located on the periphery of the network. Appendix C presents the estimated networks for the temporal moments of year and always.

Considering the *bootstrap* analyses, it can be said that the network was accurately estimated and the centrality indices remained stable (Figure 3 and Figure 4). The results for the *year* and *always* moments can be seen in Appendix D.

## 4. Discussion

The results show that the DCET is a valid and reliable multifactorial instrument for detecting depressive symptoms of all the dimensions in adults, as listed by the latest versions of the main diagnostic manuals. Although the theoretical structure comprised 20 factors, a factor analysis revealed that this number did not correspond to the actual factorisations. As indicated in the limitations of the article on the development of the DCET questionnaire [18], some of the factors had few items, which subsequently hindered the actual factorisations, in which, a loss of items left empty factors that were unable to empirically manifest the theoretical structure initially proposed. Thus, three structures of 12, 11, and 10 factors were left for the moments *month*, *year*, and *always*, respectively. In this way, the test provided an independent score in each of its factors, whose high scores corresponded to a greater presence of the specific symptomatology evaluated, so that, as Ferrando et al. [41] point out, the selected items would measure the dimensions of the depression construct.

It should be noted that thoughts of death and attention decrease coincided not only in the three time points (*month*, *year*, and *always*), but also in the explained percentage of variance, which was very similar, indicating that the items comprising them were the most representative of the construct they measure, as would be expected by the theoretical model [18]. On the other hand, the remaining factors coincided in two temporal moments, not in all three, as cases of depressed mood, appetite loss, couple deterioration, and libido diminution appeared in *month* and *year*, while anhedonia, clinical discomfort, and sleep problems appeared in *month* and *always*. It should be noted that, in the factorisation of the temporal moment *always*, the factor libido disappeared and that of couple deterioration was joined with family deterioration. This could be due perhaps to the fact that several people did not answer the items corresponding to the couple factor. As a result of this, and because the fatigue factor was formed at this time and was later eliminated for the final questionnaire, many items were eliminated in this temporal moment (see Table A3). The variance explained the decrease, although with very similar values, in the different time points measured by the DCET.

On the other hand, the network analysis helped to consolidate and clarify the proposed factorial structure and it was observed that factors such as thoughts of death, anhedonia, and depressive mood were the most central to the network, which is consistent with the initial theoretical approach to depression, and with previous works using this analysis [22,42,43]. The fact that the death thought factor was so central to the network highlights the close relationship between depression and suicide, and the importance of preventing and addressing this symptom [44,45]. However, it should be noted that these results are an approximation that should be completed with longitudinal studies to look for the predictability of relationships and the possible causality between factors [46,47].

The internal consistency analyses performed in the study sample provided adequate psychometric indicators, such as those of other questionnaires that assess depression symptoms, such as the BDI [48], the HADS [49], or the Zung Depression Scale [50], confirming one of the proposed hypotheses.

Finally, as expected, the differences found in terms of the paper and online administration formats were minimal. Given the number of items, the problems were so specific that they allowed both administrations to be unified. DIF has shown that the mode of application does not interfere with the results obtained, ensuring that the scores can be interpreted appropriately. The use of the internet in sample collection can be considered a viable option for collecting data, accessing large and heterogeneous samples, and requiring less time and financial and research staff resources [51,52]. However, the authors are aware of some of the drawbacks associated with this format, such as the difficulty in verifying the identity of the examinee and the consideration of the conditions under which it is completed.

This study has a number of limitations that should be taken into account. It is a cross-sectional study in which only the DCET was administered. The fact that no other measuring instruments were available in Spanish that contemplated the same variability of depressive symptoms, and the fact that the existing ones had little substantive coherence with the ones presented in this study, prevented us from obtaining evidence of the validity of the relationship with the other variables. Even so, future studies should focus on the convergent validity of the instrument. In addition, it should be noted that this being a self-reported measure, data on social desirability were not collected, and, consequently, this bias was not controlled. In addition, the sample had a gender disproportion that may affect the generalisability of the results. Furthermore, the sampling method was incidental and non-randomized sampling and one of the techniques used for sampling was snowball sampling, which accentuates community bias and may affect representativeness. However, the exploratory nature of this work, and the possibility of reaching a more cooperative population group from different geographical areas and not only from the university context, supported the decision to use this sampling strategy.

Given that the main objective was more focused on the analysis of the psychometric properties of the DCET, the association between depressive symptoms and socio-demographic variables was not taken into account, so that in future research, it would be convenient to analyse whether a greater or lesser presence of such symptoms is related to this type of variables. In addition, it would be convenient to compare DCET scores of clinical and non-clinical samples in order to have clinical criteria to establish the cut-off points of the questionnaire.

Regardless of the limitations, it should be noted that the sample size was large, which adds value to the data. In addition, the study contributes to deepening the multidimensional assessment of depressive symptomatology in adults. It may have practical implications in an area of current interest. Expanding the number of instruments in Spanish that allow their measurement in adults constitutes a preliminary step towards the development of research in this field that will help in the prevention of depression by aiding in screening and allowing clinicians to see progress in the treatment of depression, by comparing scores on the questionnaire. Furthermore, this will be very useful by covering a broad spectrum of depressive symptoms and separating them by areas.

A depression assessment instrument such as the one proposed here offers several advantages, among which the following stand out: (a) being an instrument that represents a construct of depression widely accepted by most researchers; (b) it allows a very exhaustive evaluation of each symptom and will help to see the treatment evolution; (c) it orients the clinician to the dimensions to be acted upon; (d) it evaluates symptoms at different times (last week, month, and more than a year ago); (e) it is compatible with the DSM and ICD diagnostic systems; (f) at a theoretical level, it allows the covariation of different factors evaluated by the test to be analysed; (g) according to expert judgment [18], the content validity of the test is adequate and adapts to the agreed definition of depression; and (h) by broadly covering depressive symptomatology, this questionnaire fits the definition of the construct.

## 5. Conclusions

In conclusion, and giving response to the aims of this study, it can be said that:

No statistically significant differences were found in terms of paper and online administration formats. The DIF has shown that the mode of application did not interfere with the results obtained, ensuring that the scores can be properly interpreted.

In response to the second objective, it can be concluded that it is a valid and reliable multifactorial instrument to detect depressive symptoms of all dimensions in adults. Three structures of 12, 11, and 10 factors have been found for the moments month, year, and always, respectively. In this way, the test provides an independent score on each of its factors.

The analysis of the internal consistency provided adequate psychometric indicators for the DCET instrument.

In the results, it was observed that depressed mood is the one that has a greater number of items as well as a greater proportion of explained variance. Furthermore, this has been confirmed with the following analysis since depressed mood plays a fundamental role in the network of depression symptoms.

## Figures and Tables

**Figure 1 ijerph-19-10788-f001:**
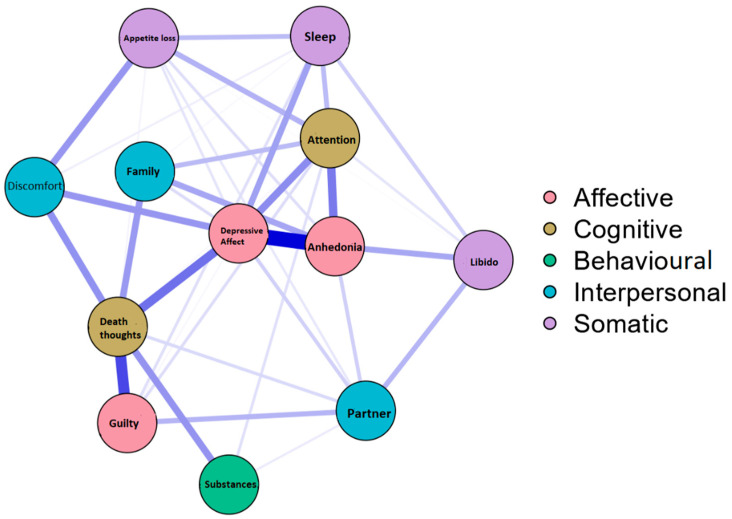
Estimated correlation network for the factors evaluated with the DCET (temporal moment ‘month’).

**Figure 2 ijerph-19-10788-f002:**
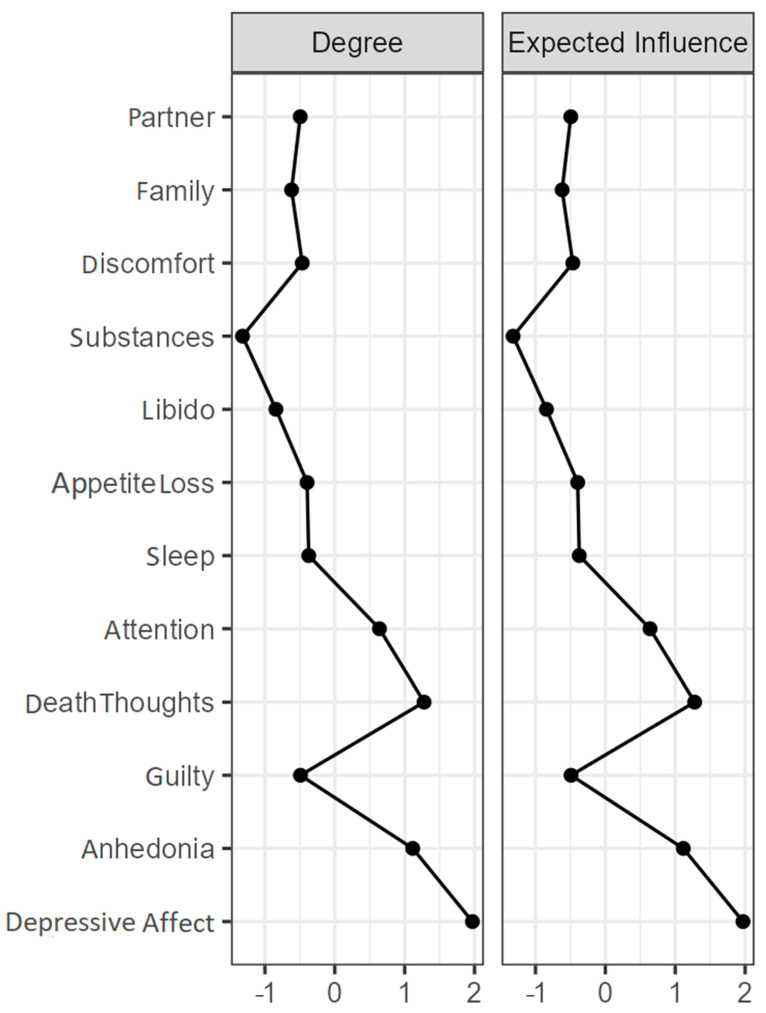
Centrality measures for the DCET factors for the moment ‘month’ (z-scores).

**Figure 3 ijerph-19-10788-f003:**
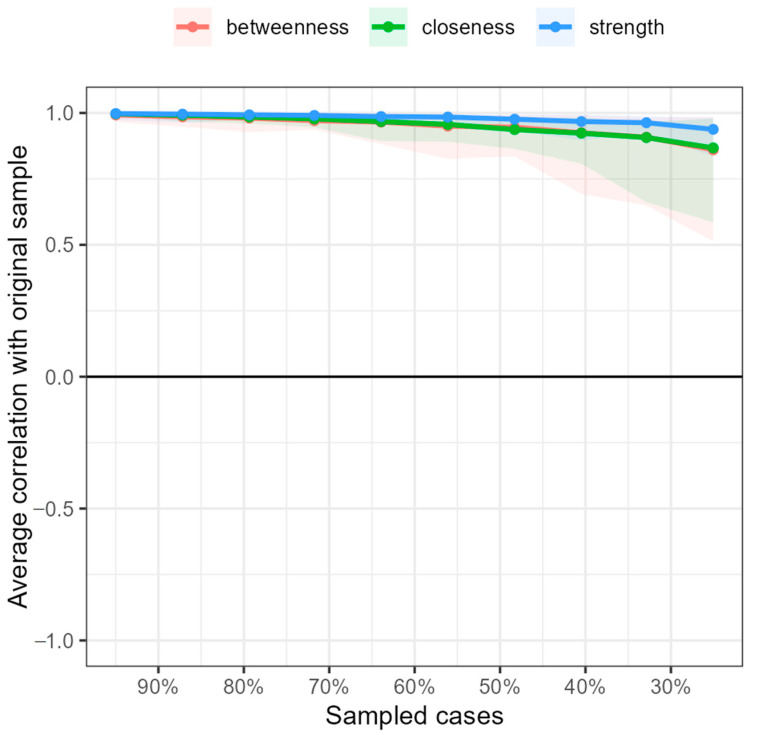
Stability of the ‘month’ moment network centrality indices with case dropping bootstrap subset. The *x*-axis indicates the percentage of cases from the original sample used in each phase. The *y*-axis shows the average correlations between the original network indices and those of the networks that were subsequently re-estimated by decreasing the increasing percentages of cases. The areas indicate a 95% CI and each line represents the correlations of betweenness, strength, and closeness.

**Figure 4 ijerph-19-10788-f004:**
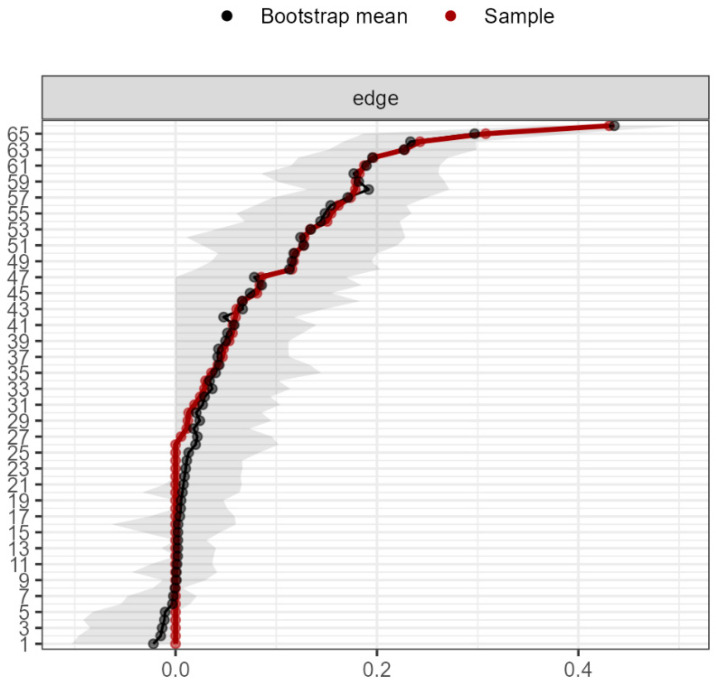
Precision in the estimation of the edges (red line, in the order of the highest to the lowest), and the 95% confidence interval of the estimates (gray line) for the network in the ‘month’ moment, estimated with the non-parametric bootstrap procedure.

**Table 1 ijerph-19-10788-t001:** Socio-demographic data of participants (N = 602).

	Online Sample (n = 500)	Paper Sample (n = 102)
Sex (female)	71.25% (357)	76.4% (78)
Age, mean (SD)	24.6 (8.72)	25.5 (6.4)
Andalusia	88% (440)	87.25% (89)
Madrid	5.2% (26)	7.84% (8)
Valencian Community and Balearic Islands	3.8% (19)	1.96% (2)
Estremadura	2.6% (13)	2.94% (3)
Galicia	0.4% (2)	
Academic level % (n)		
PhD	4.4% (22)	2.94% (3)
University Degree	76.3% (382)	38.23% (39)
University Degree in progress	6.78% (34)	40.19% (41)
High school/Baccalaureate	10.77% (54)	6.86% (7)
Vocational training	1.60% (8)	11.76% (12)
Without studies	0	0
Marital status		
Single	94.2% (472)	95% (97)
Married	4.4% (22)	3.92% (4)
Divorced	0.80%(4)	0.98% (1)
Widowed	0.40% (2)	0

## Data Availability

All data generated or analysed during this study are included in the published article. For further clarifications, the authors can be contacted at mguillot@ugr.es.

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
