# Peer review of "Network and Exploratory Factorial Analysis of the Depression Clinical Evaluation Test"

_ijerph, 2022, doi:10.3390/ijerph191710788_

Round 1
Reviewer 1 Report
Review ijerph-1849918
In their manuscript “Network and exploratory factorial analysis of the Depression Clinical Evaluation Test” Guillot-Valdés et al. examine the internal validity, DIFs and conduct factor analysis of the DECT depression assessment instrument in a mixed sample acquired by online and traditional questionnaire.
The authors present their research very clearly. Overall, the manuscript is written diligently with emphasis of possible and present limitations. Only minute corrections should be made, before this manuscript should be accepted for publication.
Minor
− Line 85 (Materials and Methods, sub header 2.2): Please add additional information about those experts. By which standards and qualifications were they selected? Did they stem from the same institution or were they gathered nationally/ internationally?
− Line 112 (Materials and Methods, sub header 2.3): Please supplement the reference number of the ethics approval.
− Lines 217 & 220 (Results, headers 3.3 & 3.4): Using the same headings for different paragraphs following, dose decrease distinguishability in the manuscript. Please rename one of the headers.
Author Response
Before starting, we want to thank the time that the reviewers have devoted to our work. In the same way we appreciate their suggestions, since these have meant a remarkable improvement of the work. Next, we answer each of the suggestions indicating the changes and the explanations. In the same way, to facilitate reading, the changes have been marked in control changes with different color.
Revisor 1st:
In their manuscript “Network and exploratory factorial analysis of the Depression Clinical Evaluation Test” Guillot-Valdés et al. examine the internal validity, DIFs and conduct factor analysis of the DECT depression assessment instrument in a mixed sample acquired by online and traditional questionnaire.
The authors present their research very clearly. Overall, the manuscript is written diligently with emphasis of possible and present limitations. Only minute corrections should be made, before this manuscript should be accepted for publication.
Minor
− Line 85 (Materials and Methods, sub header 2.2): Please add additional information about those experts. By which standards and qualifications were they selected? Did they stem from the same institution or were they gathered nationally/ internationally?
Indeed, we had not included additional information on the qualifications of the experts or their origin. The following information has been incorporated:
This instrument was created from a bank of 300 items that was subjected to a qualitative evaluation by 13 PhD experts in clinical psychology, psychometry and/or psychopathology. They had experience in the subject due to their academic training and work experience. They were gathered from different Spanish institutions/universities and contacted by email, explaining the purpose of the project and asking for their collaboration.
− Line 112 (Materials and Methods, sub header 2.3): Please supplement the reference number of the ethics approval.
The reference number was not included in case it has to be a blind information. Now we have included it (Line 125):
The study had been approved by the Human Research Ethics Committee of the University of Granada with reference 2576/CEIH/2022.
− Lines 217 & 220 (Results, headers 3.3 & 3.4): Using the same headings for different paragraphs following, dose decrease distinguishability in the manuscript. Please rename one of the headers.
We have renamed the paragraph 3.4 changing it to “Network analysis”.

Reviewer 2 Report
Thank you for giving me the opportunity to review this article “Network and exploratory factorial analysis of the Depression Clinical Evaluation Test“.
The manuscript is not ready for publication.
Major concerns:
Materials and Methods
1. According to the authors of the manuscript, a single cross-sectional study was carried out. However, as can be seen from the size of the sample selected by the Author, it was divided into „Online sample (n = 500)“ vs. „Paper sample (n =102)“. Next, on lines 325-326, the authors wrote that “Regardless of the limitations, it should be noted that the sample size is large, which adds value to the data“. Why do the authors compare 500 and 102 subjects?
2. The sampling procedure remains unclear. It must be described in this paper.
Again, on line 69, the authors revealed that “The sample consisted of 602 adults aged 18 to 85 year“. However, the average age on line 70 is stated as 24.7 ± 8.38 years. Can the authors explain these differences?
4. It was also stated that 72% of female accounted for this representative sample. Therefore, only 28% of the subjects were male. Does this mean that the questionnaire was tested on the survey for female mainly? Is this relates to the same proportion of male and female in the population, for example in Spain?
5. What about different Geographic areas of Spain the Authors stated (lines: 69-70)? Can the authors refine this data in current manuscript?
Therefore,
7. The authors should add the recruitment process of the study participants.
8. The authors should add the inclusion and exclusion criteria of this study.
9. The authors should add the methods for calculating the sample size.
10. According to the recruitment process of this study, the authors should add a flow diagram of this study.
Results
11. L 183-185: I would suggest that this information might be presented in the section of Materials and Methods (these are not the results).
Discussion
12. L 255-257: The aim of this study must be indicated not in the Discussion but in the unit of Introduction.
13. The authors wrote that “The internal consistency analyses performed in the study sample provide adequate psychometric indicators, like those of other questionnaires that assess depression, such as the BDI [48], the HADS [49] or the Zung Depression Scale [50], confirming one of the proposed hypotheses” (lines 294-296). Perhaps the authors could complement the analysis of the correlation between their prepared scale of depressive symptoms and the clinically confirmed cases of major depression.
14. The content of the author questionnaire appears to remain unclear. It seems that an accompanying 104-item file (questionnaire) should be provided.
15. L 330: The authors wrote that “...will help in the accurate diagnosis of depressive disorder, in the prediction of prognosis, and in the organization of interventions“. However, the depression disorder should be clinically confirmed by the psychiatrist. Additionally, the authors wrote only about the symptoms of depression. It is therefore necessary to supplement the existing manuscript with cut-off points for potential mental disorder symptoms associated with conditions such as "asymptomatic", "borderline", and "case-level" (in terms of the results of this study).
Conclusions
16. I would suggest that the conclusions must be rewritten. The aims of present study were namely “(1) examine the differential item functioning in two assessment procedures (online vs. paper and pencil), (2) examine the factor structure of the DCET, (3) analyze its reliability, and (4) perform a network analysis of the final factors comprising the DCET“. Thus, I would suggest that the results of the study could be generalised in the Conclusions as well as the aims of this study might be specifically answered.
Minor concerns:
1. The Abstract must be adjusted to the requirements of MDPI-IJERPH.
2. L 112: It remains unclear what the following symbols mean: “XXXXXXXXXXXXXXXXX XX XXXXXXXXXXX XXXX/XXXXXXXX/XXXXXX”.
3. L 217-220: “3.3. Reliability“ overlapes “3.4. Reliability“
4. Institutional Review Board Statement: it must be supplemented by the date of issue of the approval.
Author Response
Before starting, we want to thank the time that the reviewers have devoted to our work. In the same way we appreciate their suggestions, since these have meant a remarkable improvement of the work. Next, we answer each of the suggestions indicating the changes and the explanations. In the same way, to facilitate reading, the changes have been marked in control changes with different color.
Revisor 2:
Thank you for giving me the opportunity to review this article “Network and exploratory factorial analysis of the Depression Clinical Evaluation Test“.
The manuscript is not ready for publication.
Major concerns:
Materials and Methods
- According to the authors of the manuscript, a single cross-sectional study was carried out. However, as can be seen from the size of the sample selected by the Author, it was divided into „Online sample (n = 500)“ vs. „Paper sample (n =102)“. Next, on lines 325-326, the authors wrote that “Regardless of the limitations, it should be noted that the sample size is large, which adds value to the data“. Why do the authors compare 500 and 102 subjects?
The comparisons made between the two types of sample (online vs. paper) are to ensure that there are no significant differences with respect to the type of recruitment and confirm that their unification is possible. This analysis was only intended for the reader to understand that both samples can be unified. Still, we agree with the reviewer that it is unclear and have removed the current explanation and replaced it with the following (Line 196): “Since the sampling was done in two different formats, the first step was to confirm the absence of bias between the two applications (paper and pencil). As no serious DIF problems were observed, we proceeded to unify both samples.”
- The sampling procedure remains unclear. It must be described in this paper.
Again, on line 69, the authors revealed that “The sample consisted of 602 adults aged 18 to 85 year“. However, the average age on line 70 is stated as 24.7 ± 8.38 years. Can the authors explain these differences?
The sample was recruited mainly from university centers, co-workers, friends, relatives, etc. This population was easily accessible, due to greater internet access, ease of contact, etc. Therefore, the average age range where the sample is concentrated is 24. However, the questionnaire is indicated for people aged 18 and over, which is why it was also filled out by some older people. Hence the standard deviation of 8.38 years. It is possible that this affects the representativeness of the sample and has been added as a limitation next to the following comment (Line 340).:
“In addition, the sample had a gender and age disproportion that may affect the generalisability of the results”
- It was also stated that 72% of female accounted for this representative sample. Therefore, only 28% of the subjects were male. Does this mean that the questionnaire was tested on the survey for female mainly? Is this relates to the same proportion of male and female in the population, for example in Spain?
Although the questionnaire is designed for both sexes, in fact, the sample has a higher proportion of women. This is because, in the type of context where the sample has been collected, women are more likely to collaborate with scientific studies where questionnaires and/or surveys are applied. It is possible that this affects the representativeness of the sample and has been added as a limitation (Line 340):
“In addition, the sample had a gender and age disproportion that may affect the generalisability of the results.
- What about different Geographic areas of Spain the Authors stated (lines: 69-70)? Can the authors refine this data in current manuscript?
We have included information about the different spanish communities where the sample was taken from (see Table 1).
Therefore,
- The authors should add the recruitment process of the study participants.
In the procedure section we explain the way in which both samples were carried out, explaining the similarities and differences between them. It is possible that we have left out some relevant part, but we have gone through it thoroughly without finding anything to add. If the reviewer could explain what he thinks we need to explain better or add, we will be happy to add it.
- The authors should add the inclusion and exclusion criteria of this study.
The following paragraph has been included explaining this criteria (line 72):
“The inclusion criteria were being over 18 years old and not being diagnosed with a depressive disorder. The exclusion criteria were being under 18 years old and being diagnosed with a depressive disorder”.
- The authors should add the methods for calculating the sample size.
As no comparison was made in which the sample size could be estimated based on an effect size, this procedure was not carried out. The sample size was selected based on the analysis having the appropriate number of subjects. According to the criteria of authors such as Gorsuch (1983) or Vallejo (2013), a sample of more than 5 subjects per item is adequate. In this case, it was intended to obtain a reduction to approximately 100 items, for which a sample of 500 subjects is required.
Gorsuch, R.L. (1983). Factor analysis. Hilsdale, New Jersey: Laurence Erlbaum
Vallejo, P. M. (2013). El Análisis Factorial en la construcción e interpretación de tests, escalas y cuestionarios. Madrid: Universidad Pontificia Comillas.
- According to the recruitment process of this study, the authors should add a flow diagram of this study.
We present the flow diagram on a separate file.
Results
- L 183-185: I would suggest that this information might be presented in the section of Materials and Methods (these are not the results).
We have removed this sentence from “Results” to the “Methods” section (Line 133).
Discussion
- L 255-257: The aim of this study must be indicated not in the Discussion but in the unit of Introduction.
We removed the aim of this study from the Discussion. It is already explained in the introduction section.
- The authors wrote that “The internal consistency analyses performed in the study sample provide adequate psychometric indicators, like those of other questionnaires that assess depression, such as the BDI [48], the HADS [49] or the Zung Depression Scale [50], confirming one of the proposed hypotheses” (lines 294-296). Perhaps the authors could complement the analysis of the correlation between their prepared scale of depressive symptoms and the clinically confirmed cases of major depression.
This suggestion is very interesting but we do not have a clinical sample, as reflected in the participants section. Even so, in the limitations of the study, this reviewer's comment is presented as one of the deficiencies. (Line 352).
“In addition, it would be convenient to compare DCET scores of clinical and non-clinical samples.”
- The content of the author questionnaire appears to remain unclear. It seems that an accompanying 104-item file (questionnaire) should be provided.
The questionnaire has not been attached because it is in Spanish and has not been adapted to English. The adapted version requires a new expert opinion that is not yet available. If desired, the items can be made available by writing to the contact author.
- L 330: The authors wrote that “...will help in the accurate diagnosis of depressive disorder, in the prediction of prognosis, and in the organization of interventions“. However, the depression disorder should be clinically confirmed by the psychiatrist. Additionally, the authors wrote only about the symptoms of depression. It is therefore necessary to supplement the existing manuscript with cut-off points for potential mental disorder symptoms associated with conditions such as "asymptomatic", "borderline", and "case-level" (in terms of the results of this study).
Once again, this reviewer's comment is very appropriate, but, as we have previously mentioned, we do not have a clinical sample, so ROC curve analysis cannot be performed for this purpose. We hope that it is clear with the previously highlighted limitation included on the need for future studies that include the clinical sample.
Conclusions
- I would suggest that the conclusions must be rewritten. The aims of present study were namely “(1) examine the differential item functioning in two assessment procedures (online vs. paper and pencil), (2) examine the factor structure of the DCET, (3) analyze its reliability, and (4) perform a network analysis of the final factors comprising the DCET“. Thus, I would suggest that the results of the study could be generalised in the Conclusions as well as the aims of this study might be specifically answered.
We have already rewritten the conclusions, giving response to all the aims of the study (summarizing the results in each one). We have replaced the previous paragraph and put it after the Limitations in the Discussion section (Line 361).
Minor concerns:
- The Abstract must be adjusted to the requirements of MDPI-IJERPH.
We have adjusted the Abstract according to the MDPI-IJERPH requirements.
- L 112: It remains unclear what the following symbols mean: “XXXXXXXXXXXXXXXXX XX XXXXXXXXXXX XXXX/XXXXXXXX/XXXXXX”.
The reference number of the ethics committee was not included in case it has to be a blind information for the review process. Now we have included it.
- L 217-220: “3.3. Reliability“ overlapes “3.4. Reliability“
We have renamed the paragraph 3.4 changing it to “Network analysis”.
- Institutional Review Board Statement: it must be supplemented by the date of issue of the approval.
The date has been included.

Round 2
Reviewer 2 Report
The authors have changed this manuscript and I congratulate them on that. However, a large number of uncertainties and unaddressed issues remain. My further major and minor concerns are as follows:
Major concerns:
1. Line 262: The authors wrote: “...depressive disorder“, “depression“. The current questionnaire form is and can only be used for screening purpose in order to detect potential health disorders or diseases in people who do not have any symptoms of disease. However, both depressive disorder and depression can only be identifed as well as diagnosed with the help of a psychiatrist or psychotherapist. It seems necessary to write, for example, “depressive symptoms”, “quality of depressive symptoms” throughout the manuscript.
2. Line 302: The authors wrote: “Finally, as expected, no statistically significant differences were found in terms of the paper and online administration formats”. It seems necessary to specify statistical significance (p-value) in the section namely Methods and Materials (line: 75, Table 1).
3. Line 330: The authors wrote: “…it should be noted that the sample size is large, which adds value to the data“. However, the sample size must be representative too. Therefore, my previous comments are still valid:
The sampling procedure remains unclear (what was the method of sampling strategy: 1) simple random, 2) stratified random, 3) cluster, and 4) systematic?). It must be described in this paper. The authors should add the recruitment process of the study participants. The authors should add the methods for calculating the sample size (sample size must be calculated; I recommend online calculator: https://www.openepi.com/SampleSize/SSCohort.htm). According to the recruitment process of this study, the authors should add a flow diagram of this study. In order to achieve this, at first the authors must select a target population followed by a study population (identification pahse). Further, in accordance with the inclusion and exclussion criteria, select the proportion of the population (inclusion phase) should be analysed (analysis phase). I would also think that your target population is all Spanish residents. The flowchart must be corrected. There is certainly a lot of literature on this issue. Information may also be accessed: https://www.strobe-statement.org/.
4. Line 335: The authors wrote: “...will help in the accurate diagnosis of depressive disorder, in the prediction of prognosis, and in the organization of interventions“. Dear Authors, The similar scales are developed for both early prevention and revealing only the severity of potential depressive symptoms (it is not a diagnosis). However, the diagnosis is performed by psychiatrist as well as treatment is administered only by him. The entire paper must be “amortised” in a such a case.
5. Line 353: The authors write about the global points of their scale and their interpretation. Thus, it remains unclear how to interpret the severity of depressive symptoms accurately, taking into account cut-off points. In what specific case is it necessary to recommend a consultation of psychologist/psychotherapist (taking into account the sum of the scores summed up on the Author's scale) to the prospective audiences after the screening? How does the range of these scores relate to the severity of the depressive symptoms?
6. Line 361: The authors wrote: “…Depressed mood play a fundamental role in the network of depression symptoms”. What does this conclusion mean? Why is it justified?
7. The next authors wrote that “Giving special attention to these symptoms can be essential to reduce the risk of developing a depressive disorder“. How can the authors explain this global effect? Will focusing on symptoms really reduce the risk of developing depression? It seems that this information should be further examined in both the introduction and discussion sections of this manuscript.
Minor concerns:
1. Since the manuscript itself has the direction of biomedical sciences, I suggest that the entire Article be corrected using the past tenses (such common rules exist).
2. The manuscript text contains a large number of words that start with uppercase letters in the middle of text sentences. I propose that this be brought into line with the requirements.
3. I suggest that authors correct words such as “analyze“ (lines: 13, 61, 65, 207), “organization“ (line 28), “generalized“ (line 45), “factorization“ (lines 135, 202, 205, 221..), “regularization“ (line 161) for English dictionary values throughout the entire manuscript. For example, analyse, generalised and etc.
4. Line 190: “R2Nagelkerke“ maybe could be corrected to “Nagelkerke R2“.
5. Table 1: The units of measurement '% (n)' accompanied by variables appear to be missing.
6. Table 1: the authors write about “gender analysis“. I recommend that “gender” must be changed to “sex”. Please see the section “Sex and Gender in Research“ (available online: https://www.mdpi.com/journal/ijerph/instructions).
7. Line 201: The authors wrote „The Bartlett's test of sphericity was statistically significant“ (so, test or results?).
8. The manuscript contains many unclear terms for future readers, which makes it necessary for the authors to produce an acronym table too.
Author Response
The authors would like to thank again the comments provided by the reviewer, especially considering the speed with which he/she has done it. We appreciate the effort to improve our work without delaying times. We hope in this review to adequately answer the suggestions and comments proposed. We go on to list the changes made (with change control) or to comment on the issues that cannot be changed.
Major concerns:
- Line 262: The authors wrote: “...depressive disorder“, “depression“. The current questionnaire form is and can only be used for screening purpose in order to detect potential health disorders or diseases in people who do not have any symptoms of disease. However, both depressive disorder and depression can only be identifed as well as diagnosed with the help of a psychiatrist or psychotherapist. It seems necessary to write, for example, “depressive symptoms”, “quality of depressive symptoms” throughout the manuscript.
We have made the requested changes to the term “depressive disorder” and changed it to “depressive symptoms” throughout the manuscript, where applicable.
- Line 302: The authors wrote: “Finally, as expected, no statistically significant differences were found in terms of the paper and online administration formats”. It seems necessary to specify statistical significance (p-value) in the section namely Methods and Materials (line: 75, Table 1).
DIF analyses were performed on all the items in three different response versions. There were only DIF problems on a couple of items in one of the response options. Although this is correctly developed in the results, it is true that in the discussion, it may be doubtful for the reader. It has been replaced with the following sentence (Line 308):
“Finally, as expected, the differences were found in terms of the paper and online administration formats were minimal. Given the number of items, the problems found are so specific that they allow both administrations to be unified. “
- Line 330: The authors wrote: “…it should be noted that the sample size is large, which adds value to the data“. However, the sample size must be representative too. Therefore, my previous comments are still valid:
The sampling procedure remains unclear (what was the method of sampling strategy: 1) simple random, 2) stratified random, 3) cluster, and 4) systematic?). It must be described in this paper. The authors should add the recruitment process of the study participants. The authors should add the methods for calculating the sample size (sample size must be calculated; I recommend online calculator: https://www.openepi.com/SampleSize/SSCohort.htm). According to the recruitment process of this study, the authors should add a flow diagram of this study. In order to achieve this, at first the authors must select a target population followed by a study population (identification pahse). Further, in accordance with the inclusion and exclussion criteria, select the proportion of the population (inclusion phase) should be analysed (analysis phase). I would also think that your target population is all Spanish residents. The flowchart must be corrected. There is certainly a lot of literature on this issue. Information may also be accessed: https://www.strobe-statement.org/.
We thank the reviewer for the website as it has many resources that will surely serve us from here on out. However, as we indicated in the previous review, in this case a sample size estimator cannot be used since these analyzes do not present an effect size on which to base it. For this reason, we have used another sample size calculator, QuestionPro 2022 (https://www.questionpro.com/es/calculadora-de-muestra.html). Here, the recommended sample size at 95% and with a margin of error of 3.99 is 604. This is based on the sample size formula recommended by other authors (Chiaran & Biswan, 2013).
Regarding the diagram, it has been modified according to the STROBE criteria.
Finally, regarding the sampling, as mentioned in the article, it was incidental and not randomized. We understand that this supposes a serious limitation and so is indicated in line 325:
“Furthermore, the sampling method was incidental and non-randomized sampling and one of the techniques used for sampling was snowball sampling, which accentuates community bias and may affect representativeness."
- Line 335: The authors wrote: “...will help in the accurate diagnosis of depressive disorder, in the prediction of prognosis, and in the organization of interventions“. Dear Authors, The similar scales are developed for both early prevention and revealing only the severity of potential depressive symptoms (it is not a diagnosis). However, the diagnosis is performed by psychiatrist as well as treatment is administered only by him. The entire paper must be “amortised” in a such a case.
We have changed the mentioned sentence and wrote this (Line 342), in order to clarify this point:
“… will help in the prevention of depression by aiding in screening and allowing clinicians to see progress in the treatment of depression, by comparing scores on the questionnaire. Furthermore, this will be very useful by covering a broad spectrum of depressive symptoms and separating them by areas”.
- Line 353: The authors write about the global points of their scale and their interpretation. Thus, it remains unclear how to interpret the severity of depressive symptoms accurately, taking into account cut-off points. In what specific case is it necessary to recommend a consultation of psychologist/psychotherapist (taking into account the sum of the scores summed up on the Author's scale) to the prospective audiences after the screening? How does the range of these scores relate to the severity of the depressive symptoms?
We have changed the sentence in Line 351 and we wrote this:
“It allows a very exhaustive evaluation of each symptom and will help to see the treatment evolution”
Indeed, it would be relevant to have a clinical population or clinical criteria. However, such information is not available. Currently, the lack of patients is included as a limitation and it has been expanded by saying that there are no clinical criteria to establish the cut-off point (Line 336).
- Line 361: The authors wrote: “…Depressed mood play a fundamental role in the network of depression symptoms”. What does this conclusion mean? Why is it justified?
Indeed this fact may not be well explained and has been replaced by this:
“In the results, it has been observed that depressed mood is the one that has a greater number of items as well as a greater proportion of explained variance. Furthermore, this has been confirmed with the following analysis since depressed mood play a fundamental role in the network of depression symptoms.”
- The next authors wrote that “Giving special attention to these symptoms can be essential to reduce the risk of developing a depressive disorder“. How can the authors explain this global effect? Will focusing on symptoms really reduce the risk of developing depression? It seems that this information should be further examined in both the introduction and discussion sections of this manuscript.
In effect, this phrase is not based on the results obtained and has been eliminated.
Minor concerns:
- Since the manuscript itself has the direction of biomedical sciences, I suggest that the entire Article be corrected using the past tenses (such common rules exist).
We have reviewed the entire article and we have changed the verbs that were not in past tenses.
- The manuscript text contains a large number of words that start with uppercase letters in the middle of text sentences. I propose that this be brought into line with the requirements.
We have changed this in the case of the facets throughout the entire manuscript (depression, anhedonia, thoughts of death, etc). We hope now it is in line with the requirements.
- I suggest that authors correct words such as “analyze“ (lines: 13, 61, 65, 207), “organization“ (line 28), “generalized“ (line 45), “factorization“ (lines 135, 202, 205, 221..), “regularization“ (line 161) for English dictionary values throughout the entire manuscript. For example, analyse, generalised and etc.
We have corrected this words for English dictionary values throughout the manuscript; we have changed them to analyse, generalised, organisation, factorisation and regularisation.
- Line 190: “R2Nagelkerke“ maybe could be corrected to “Nagelkerke R2“.
We have corrected this.
- Table 1: The units of measurement '% (n)' accompanied by variables appear to be missing.
This was referred to the next columns, where we present first the percentage and then the “n” of women both in online and paper simple. However, we have removed this ‘%(n)’ so that there is no confussion for the reader.
- Table 1: the authors write about “gender analysis“. I recommend that “gender” must be changed to “sex”. Please see the section “Sex and Gender in Research“ (available online: https://www.mdpi.com/journal/ijerph/instructions).
We have replaced “Gender” with “Sex”.
- Line 201: The authors wrote „The Bartlett's test of sphericity was statistically significant“ (so, test or results?).
We have changed the sentence in Line 201, including this:
“The results of the Bartlett's test of sphericity was statistically significant (p < 0.001) in all cases”.
- The manuscript contains many unclear terms for future readers, which makes it necessary for the authors to produce an acronym table too.
We have included an acronym table (Line 391).